

# Tumor marker elevations in chronic kidney disease: a retrospective study

Liuxia You[1,*], Zhongying Xu[2,*], Qiaoling Liu[1], Guoming Jin[3] and Changting Wang[4]

[1] Department of Clinical Laboratory, The Second Affiliated Hospital of Fujian Medical University, Quanzhou, Fujian, China

[2] Department of Emergency and Critical Care Medicine, The Affiliated Suzhou Hospital of Nanjing Medical University, Suzhou Municipal Hospital, Gusu School of Nanjing Medical University, Suzhou, Jiangsu, China

[3] Department of Internal Medicine, Guli People's Hospital, Suzhou, Jiangsu, China

[4] Department of General Surgery, The Second Affiliated Hospital of Fujian Medical University, Quanzhou, Fujian, China

[*] These authors contributed equally to this work.

Corresponding authors
Guoming Jin, ben22221111@163.com
Changting Wang, changtingw@sina.com

## ABSTRACT

**Aim**. To investigate the changes in tumor markers such as carbohydrate antigen 125 (CA125), carbohydrate antigen 153 (CA15-3), carbohydrate antigen 19-9 (CA19-9), carbohydrate antigen 72-4 (CA72-4), pro-gastrin-releasing peptide (proGRP), and human epididymis protein 4 (HE4) in patients with chronic kidney disease (CKD) as compared to healthy individuals, and to analyze the potential indicators that can be utilized for monitoring early renal dysfunction, was the aim of this study.

**Methods**. This retrospective study involved 522 participants from our hospital, including 172 patients with CKD and 350 controls. CKD patients were divided into CKD1, CKD2, CKD3, CKD4, and CKD5 stages according to the estimated glomerular filtration rate (eGFR). Plasma creatinine and general clinical data were collected upon admission.

**Results**. The level of tumor biomarkers, including carcinoembryonic antigen (CEA), CA125, CA15-3, CA19-9, NSE, proGRP, and HE4, were elevated in patients with CKD ($p < 0.05$). There were differences in CEA, CA125, CA19-9, NSE, proGRP, and HE4 between the control and CKD groups. The subgroup study showed that CEA, CA15-3, CA19-9, NSE, proGRP, and HE4 were elevated in patients with CKD having normal serum creatinine compared with those in the control group ($p < 0.05$). ProGRP and HE4 have high predictive values for early renal insufficiency with area under the curve of 0.736 and 0.931, respectively ($p < 0.05$). Furthermore, HE4 and proGRP were positively correlated with the stages of CKD, with correlation coefficients of 0.623 and 0.712, respectively.

**Conclusions**. Patients with CKD have higher tumor markers, some of which are helpful for the early diagnosis of renal impairment.

## INTRODUCTION

In recent years, chronic kidney disease (CKD) has attracted increasing attention as a public health issue because of the serious economic burden it can impose on patients and

their families. In addition to the primary kidney disease, CKD is often accompanied by hypertension and diabetes, which are high-risk factors for cardiovascular events. Therefore, cardiovascular events are also considered a CKD-related social burden besides kidney replacement therapy (*Bello, Nwankwo & El Nahas, 2005*). According to a cross-sectional study, the incidence rate of CKD has reached 10.8%, and the total number of patients suffering from CKD is 119.5 million (*Zhang et al., 2012*). A similar overall prevalence rate of CKD has also been reported in other countries and regions (*GBD Chronic Kidney Disease Collaboration, 2020*).

Lung cancer is one of the most serious threats to human health, with 2.38 million new cases and 1.27 million deaths annually in the USA (*Siegel et al., 2023*). Most patients with lung cancer are diagnosed in the late stage due to the lack of typical symptoms. Hence, the early diagnosis of lung cancer can have very meaningful implications. Tumor markers refer to substances produced by tumor cells that exist in the tumor tissue or are secreted into the tumor. Furthermore, it can be produced by host cells due to tumor cell stimulation. Among the early diagnostic methods for lung cancer, tumor markers have attracted much attention. Owing to their non-invasive characteristics, tumor markers, such as carbohydrate antigen 125 (CA125), carbohydrate antigen 153 (CA15-3), carbohydrate antigen 19-9 (CA19-9), carbohydrate antigen 72-4 (CA72-4), pro-gastrin-releasing peptide (proGRP), and human epididymis protein 4 (HE4), have been widely used in the diagnosis, monitoring, and evaluation of tumors. Molina reported that the combination of tumor markers, including CEA, SCC, CY211, CA15-3, NSE, and ProGRP can achieve an area under the curve (AUC) of 0.85 in the diagnosis of lung cancer (*Molina et al., 2016*).

Owing to metabolic changes in patients with CKD, the commonly used tumor markers are often abnormal in these patients (*Miao et al., 2022*; *Huang et al., 2023*). Hence, there is considerable controversy over the application of tumor markers in patients with renal insufficiency. Moreover, patients with early renal dysfunction are mostly asymptomatic; however, patients possess significantly impaired renal function when they are symptomatic. Therefore, the early detection of renal function damage, particularly in the CKD1 and CKD2 phases, is beneficial because it can facilitate the implementation of the necessary measures at an early stage to slow down the deterioration of renal function, prolong the progression to end-stage renal disease, and thus, improve patients' quality of life. We also investigated the changes of multiple traditional tumor markers, including CA125, CA15-3, CA19-9, CA72-4, NSE, proGRP, and HE4 in patients with CKD compared with healthy individuals. In addition, we analysed the potential indicators that can be used for monitoring early renal dysfunction.

## MATERIAL AND METHODS

### Patients

A total of 522 participants were recruited from the second affiliated hospital of Fujian Medical University from January 2021 to January 2022, including 172 patients with CKD and 350 controls (Fig. 1). Pregnant and lactating women, as well as patients with related tumors, liver cirrhosis, skin diseases, severe cardiovascular and cerebrovascular diseases,

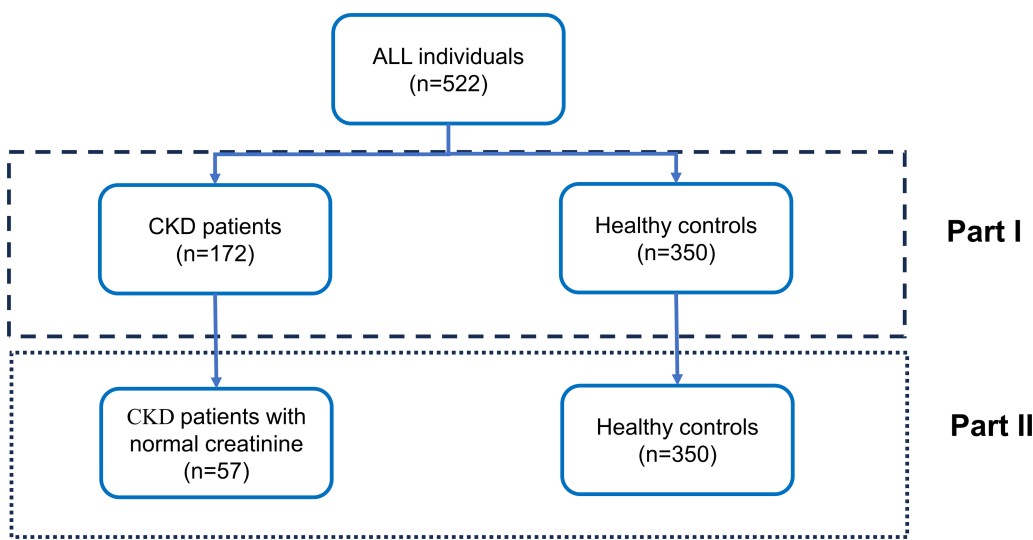

**Figure 1  Overall study design.** The study cohort comprised 522 participants, stratified into two distinct groups in Part I: individuals diagnosed with chronic kidney disease (CKD) ($n = 172$) and healthy control subjects ($n = 350$). In Part II, a subset of CKD patients exhibiting normal creatinine levels ($n = 57$) was selected from the initial CKD cohort, while the healthy control group ($n = 350$) remained constant for comparative analysis. The design is delineated into two analytical phases, as demarcated by the dashed box representing Part I and the dotted box representing Part II.

severe infections, pleural and ascites fluids and benign gynecological disorders, were excluded from the study. This study protocol was approved by the Ethics Committee of the Second Affiliated Hospital of Fujian Medical University, and informed consent was obtained from the patients who signed the consent form. (Ethical number:[2021] Fujian Second Affiliated Hospital Ethics Approval No. (399).

The epidemiology collaboration (CKD-EPI) equation was used to calculate the estimated glomerular filtration rate (eGFR) (*Levey et al., 2009*). All patients were divided into the CKD1, CKD2, CKD3, CKD4, and CKD5 stages according to their eGFR values. The overall study design was displayed in Fig. 1. As shown in the figure, our study is divided into two parts. Initially, we analyzed the potential of tumor marker concentration changes to differentiate between CKD patients and healthy controls. Subsequently, we further assessed the ability of these changes in tumor marker concentrations to distinguish between CKD patients with normal creatinine and healthy controls.

## Assay for tumor markers

The serum concentrations of biomarkers were measured with the commercially available Roche immunoassay (Cobas E601; Roche Diagnostics, Basel, Switzerland), following the defined protocol. The respective reagent catalog numbers are listed below: CEA (Catalog No. 11731629322), proGRP (Catalog No. 09007636190), HE4 (Catalog No. 05950929190), NSE (Catalog No. 12133113122), CA199 (Catalog No. 11776193122), CA153 (Catalog No. 03045838122) and CA125 (Catalog No. 11776223190).

## Statistical analysis

SPSS (*v.* 20) was used for data management and statistical analysis. Counting data were expressed in frequency and percentage. To ensure the accuracy of data presentation, we conducted normality tests on the relevant variables before reporting. Data that follow a normal distribution are presented as the mean ± standard deviation, whereas data that do not follow a normal distribution are presented using the interquartile range (IQR). Furthermore, the *t*-test or Chi-squared test was used for comparisons between the two groups. Specifically, the *t*-test was applied when the data followed a normal distribution with homogeneity of variances, while non-parametric tests were employed when the data did not meet these assumptions. For categorical variables, the Chi-squared test was used. For comparisons among multiple groups, the Kruskal–Wallis test was utilized.

In our analysis of the relationship between the concentrations of tumor marker and the staging of CKD patients, we treated CKD staging as an ordinal variable. The Spearman correlation analysis was used to determine the relationship between tumor markers and the different stages of CKD. We used univariate analysis to assess the discriminative ability of various tumor markers for identifying CKD patients. $p < 0.05$ was considered statistically significant.

# RESULTS

## Clinical characteristics of patients and control

The general clinical characteristics of patients and controls are summarised in Table 1. A total of 522 patients were included in the study, including 172 patients with CKD and 350 controls. The proportion of men in the CKD group was slightly higher than that in the control group. In addition, the individuals in the CKD group were older than those in the control group. The common complications of CKD included hypertension (29.65%), type 2 diabetes (15.12%), hyperuricemia (13.37%) and anaemia (11.63%). Urinary protein was abnormal in 45.35% of patients with CKD. In addition, the concentrations of tumor biomarkers, including CEA, CA125, CA15-3, CA19-9, NSE, proGRP, and HE4, were elevated in patients with CKD. There were differences in CEA, CA125, CA19-9, NSE, proGRP and HE4 between the control group and patients in the different stages of CKD (Table 2).

## Correlation analysis between tumor markers and different stages of patients with CKD

Our previous analysis suggested that the concentrations of some tumor markers were elevated in patients with CKD. Spearman correlation analysis was used to determine the correlation between the concentrations of tumor markers and the different stages of CKD. Based on the correlation coefficients and *p*-values, CEA, CA15-3 and CA125 show a relatively low correlation with the staging of CKD patients. In contrast, proGRP and HE4 exhibit a positive correlation with the staging of CKD, with correlation coefficients of 0.712 and 0.623, respectively (Fig. 2).

**Table 1  Clinical characteristics**

|  | Control group (n = 350) | CKD group (n = 172) | p |
|---|---|---|---|
| Gender Male (%) | 207 (59.14) | 119 (69.19) | 0.026 |
| Age (years) | 38.19 (32.00, 45.00) | 57.01 ± 14.99 | <0.001 |
| Stage of CKD (%) |  |  |  |
| 1 |  | 38 (22.09) |  |
| 2 |  | 36 (20.93) |  |
| 3 |  | 38 (22.09) |  |
| 4 |  | 22 (12.79) |  |
| 5 |  | 38 (22.09) |  |
| Comorbidity & Complication |  |  |  |
| Hypertension (%) |  | 51 (29.65) |  |
| Diabetes (%) |  | 26 (15.12) |  |
| Cardiovascular diseases (%) |  | 9 (5.23) |  |
| Hyperuricemia/Gout (%) |  | 23 (13.37) |  |
| Anemia (%) |  | 20 (11.63) |  |
| Ur (mmol/L) | 4.78 ± 1.02 | 11.43 (5.69, 13.85) | <0.001 |
| Cr (umol/l) | 70.06 (58.22, 81.85) | 225.26 (83.00, 357.10) | <0.001 |
| Urine Protein (%) |  |  | <0.001 |
| − | 346 (98.86) | 94 (54.65) |  |
| + | 0 (0.0) | 25 (14.53) |  |
| ++ | 0 (0.0) | 24 (13.95) |  |
| +++ | 0 (0.0) | 29 (16.86) |  |
| ++++ | 0 (0.0) | 0 (0.0) |  |
| CEA (ng/mL) | 1.70 (1.01, 2.11) | 3.18 (1.85, 3.82) | <0.001 |
| CA125 (U/mL) | 12.34 (8.09, 14.62) | 41.98 (11.30, 43.97) | <0.001 |
| CA15-3 (U/mL) | 9.89 ± 4.91 | 13.77 ± 8.23 | 0.003 |
| CA19-9 (U/mL) | 9.60 (5.28, 11.74) | 27.55 (7.25, 27.58) | <0.001 |
| CA72-4 (U/mL) | 4.31 (1.10, 3.96) | 7.11 (0.89, 1.45) | 0.028 |
| NSE (ng/mL) | 11.88 (10.10, 13.22) | 15.19 ± 4.62 | <0.001 |
| proGRP (pg/mL) | 40.72 ± 10.28 | 112.61 (56.09, 169.10) | <0.001 |
| HE4 (pmol/L) | 39.43 ± 8.39 | 313.97 (82.22, 429.80) | <0.001 |

**Notes.**

Data that follow a normal distribution are presented as the mean ± standard deviation, whereas data that do not follow a normal distribution are presented using the interquartile range.

CKD, chronic kidney diseases; Ur, urea nitrogen; Cr, creatinine; CEA, carcinoembryonic antigen; CA125, cancer antigen 125; CA15-3, cancer antigen 15-3; CA19-9, carbohydrate antigen 19-9; CA72-4, cancer antigen 72-4; NSE, neuron specific enolase; proGRP, pro-gastrin-releasing peptide; HE4, Human epididymis protein 4.

## Value of tumor markers in predicting renal dysfunction

Based on previous analysis, we observed that some tumor markers, including CEA, CA125, CA15-3, CA19-9, proGRP, and HE4, were elevated in patients with CKD. We further analysed the value of tumor indicators in predicting renal insufficiency. As shown in Fig. 3, HE4 and proGRP showed the best diagnostic performance, with an AUC of 0.935 and 0.890, respectively (Fig. 3).

You et al. (2025), *PeerJ*, DOI 10.7717/peerj.19240

**Table 2  The level of biomarkers in control group and different stage of CKD patients-1.**

|  | CEA | CA125 | CA15-3 | CA19-9 | NSE | proGRP | HE4 |
|---|---|---|---|---|---|---|---|
| Control | 1.54 (343; 1.0, 2.11) | 10.96 (242; 8.09, 14.62) | 8.78 (143; 6.13, 13.23) | 7.76 (340; 5.18, 11.72) | 11.54 (323; 10.10, 13.22) | 39.11 (350; 33.23, 47.61) | 38.85 (117; 33.26, 43.00) |
| CKD 1 | 2.63 (38; 2.04, 3.58) | 11.48 (6; 9.32, 20.82) | 10.65 (6; 8.77, 15.85) | 9.28 (21; 5.86, 13.33) | 12.74 (5; 11.30, 18.86) | 47.52 (37; 35.42, 72.89) | 76.95 (7; 57.42, 88.55) |
| CKD 2 | 2.14 (35; 1.52, 3.58) | 11.62 (10; 5.82, 32.83) | 10.19 (10; 5.98, 20.15) | 10.855 (28; 5.98, 18.46) | 13.85 (7; 9.65, 17.65) | 55.44 (36; 46.31, 71.28) | 82.98 (15; 47.84, 230.50) |
| CKD 3 | 3.06 (35; 2.00, 4.08) | 23.43 (13; 14.12, 68.82) | 12.16 (13; 8.48, 23.11) | 11.045 (32; 6.86, 37.580) | 14.75 (3; 9.65, 18.03) | 87.52 (38; 64.58, 101.41) | 187.00 (13; 127.95, 440.80) |
| CKD 4 | 2.36 (21; 1.98, 4.53) | 77.16 (5; 9.88, 147.45) | 11.31 (4; 6.22, 13.05) | 12.67 (18; 5.87, 29.27) | 15.19 (5; 12.04, 18.86) | 140.95 (22; 91.52, 192.78) | 352.65 (6, 199.92, 727.27) |
| CKD 5 | 3.01 (38; 2.06, 3.88) | 17.02 (12; 12.49, 39.95) | 13.48 (11; 9.73, 16.84) | 21.00 (35; 10.46, 30.86) | 15.54 (9; 12.88, 20.43) | 193.8 (38; 140.45, 236.45) | 522.80 (11, 309.20, 630.10) |
| $H$ | 101.268 | 31.492 | 10.358 | 35.040 | 18.402 | 254.874 | 83.353 |
| $p$ | <0.001 | <0.001 | 0.066 | <0.001 | 0.002 | <0.001 | <0.001 |

**Notes.**

Data are presented as median (number of patients; interquartile range).
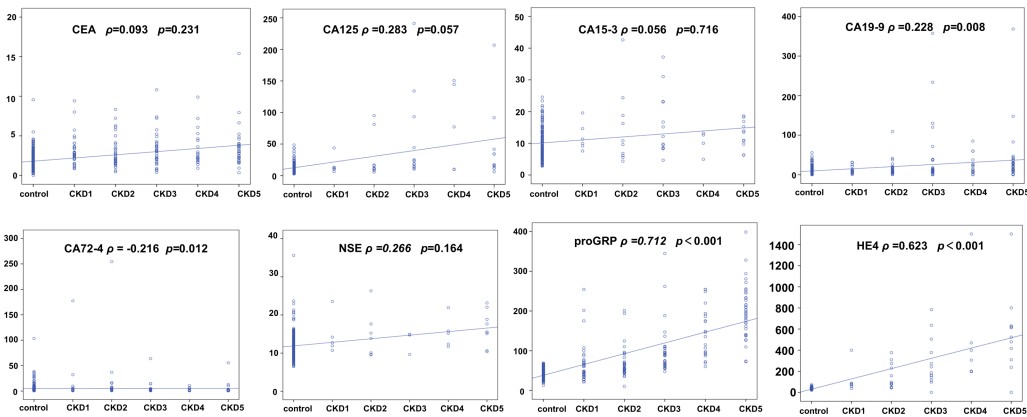

**Figure 2  Correlation analysis of tumor markers across different stages of chronic kidney disease (CKD).** The Spearman correlation analysis of tumor markers and CKD stages, comparing them to the control group. Each plot displays the correlation between a tumor marker and CKD stage, with Spearman's rho and *p*-value indicated. Markers include CEA, CA125, CA15-3, CA19-9, CA72-4, NSE, proGRP, and HE4.

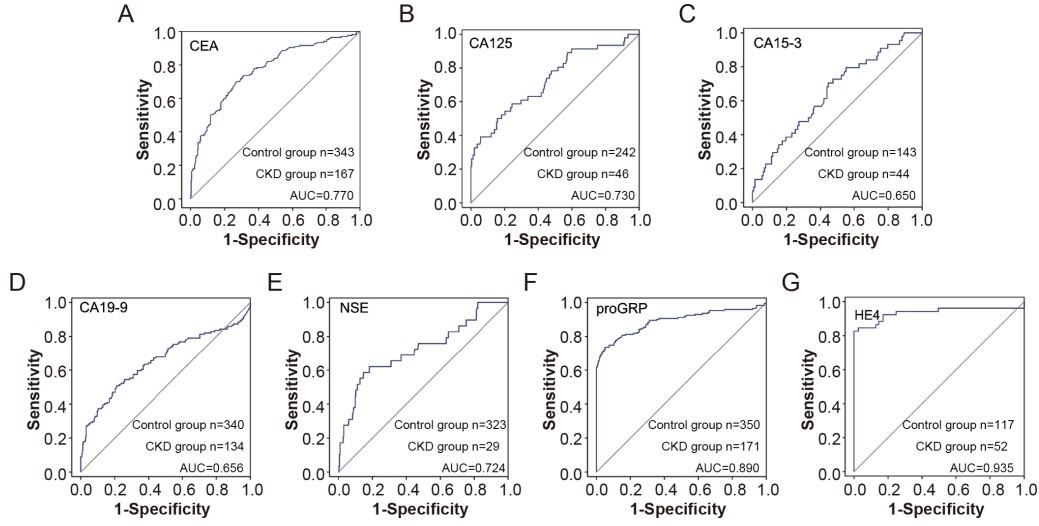

**Figure 3  The efficiency of tumor markers in distinguish CDK patients from healthy control.** The receiver operating characteristic (ROC) curves for various tumor markers in distinguishing chronic kidney disease (CKD) patients from healthy controls. The markers include CEA (A), CA125 (B), CA15-3 (C), CA19-9 (D), NSE (E), proGRP (F), and HE4 (G). The sensitivity and 1-specificity values are plotted for each marker, with the diagonal line representing the reference line of no discrimination. The area under the curve (AUC) for each marker indicates its diagnostic efficiency.

Furthermore, the subgroup results showed that compared with the control group, CEA, CA15-3, CA19-9, NSE, proGRP and HE4 were elevated in patients with CKD with normal serum creatinine, and the differences were statistically significant, suggesting that the elevation of tumor indicators are conducive to the early diagnosis of CKD. Furthermore,

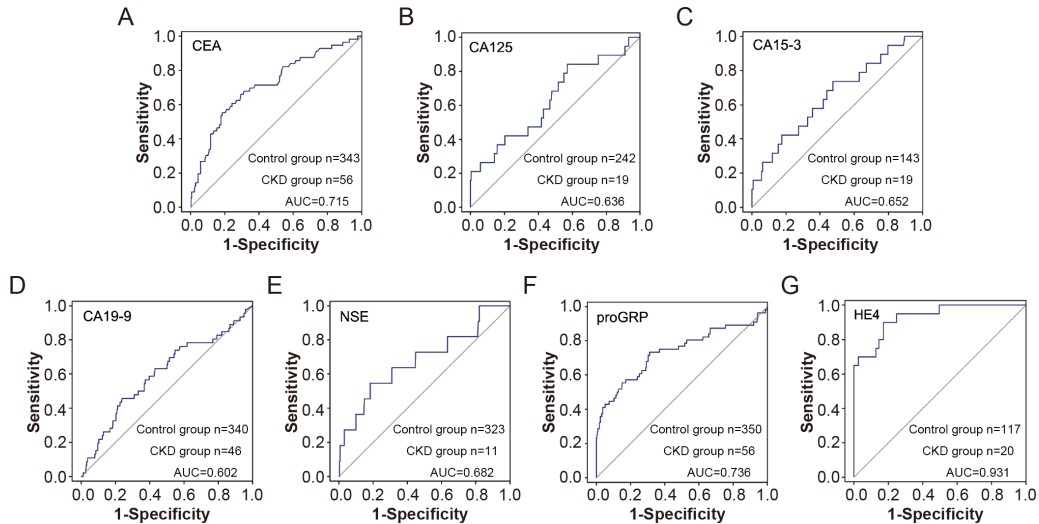

**Figure 4** The efficiency of tumor markers in distinguish CDK patients with normal serum creatinine from healthy controls. The receiver operating characteristic (ROC) curves for various tumor markers in distinguishing chronic kidney disease (CKD) patients with normal serum creatinine from healthy controls. The markers include CEA (A), CA125 (B), CA15-3 (C), CA19-9 (D), NSE (E), proGRP (F), and HE4 (G). The sensitivity and 1-specificity values are plotted for each marker, with the diagonal line representing the reference line of no discrimination. The area under the curve (AUC) for each marker indicates its diagnostic efficiency.

proGRP and HE4 showed high predictive values for early renal insufficiency with AUC of 0.736 and 0.931, respectively (Fig. 4).

## DISCUSSION

Tumor biomarkers refer to substances that are abnormally produced by malignant tumor cells or hosts in response to tumor stimulation. These markers originate from the synthesis and secretion of normal tissues (*Wang et al., 2023*) and are used to reflect the occurrence and development of the tumor and monitor its response to treatment (*Li et al., 2018*). Tumor markers exist in the tissues, body fluids, and excreta of patients with tumors and can be detected using immunological, biological, and chemical methods. In addition to being associated with tumors, tumor markers are also associated with pregnancy, inflammation, liver function, and renal dysfunction (*Estakhri et al., 2013*).

The incidence rate of CKD in China has reached approximately 11%, which is a serious threat to human health. The serum tumor marker concentration depends on the balance between the rate of tumor marker production and the metabolic rate in the body. The changes in tumor marker concentrations in patients with CKD remain controversial, with some studies showing no change and others suggesting increased concentrations of CA 15-3, CEA, and CA 19-9 in patients with CKD (*Amiri, 2016*). For instance, a study by *Tong et al. (2013)* evaluated tumor markers such as CEA, NSE, AFP, CY211, and SCC in CKD patients. The study included 539 non-dialysis CKD patients and 223 healthy controls. It found that the false-positive rates of CEA, SCC, and CY211 were significantly higher in

CKD patients compared to healthy controls. There was no significant difference between CKD patients and healthy controls for AFP and NSE. Additionally, the false-positive rates of Cyfra21-1 and SCC in CKD patients increased progressively with the worsening of CKD. *Rani et al. (2019)* showed that CEA, HCG, CA19-9, and CA15-3 concentrations in patients with CKD were significantly higher than those in the control group. There was no difference in tumor marker concentrations between stage 4 and stage 5 CKD. This may be due to impaired excretion owing to metabolic changes and reduced GFRs in patients with CKD (*Rani et al., 2019*). Furthermore, *Huang et al. (2023)* evaluated the correlation of serum Galectin-3 and CA15-3 concentrations with kidney function, revealing a significant negative correlation between Galectin-3, CA15-3, and eGFR. This relationship was more pronounced in overweight and obese individuals. These studies suggest that tumor marker concentrations in CKD populations may be influenced by kidney function. However, the existing literature has not further evaluated the role of abnormal tumor marker concentrations in the differentiation of CKD, especially in the identification of early-stage CKD with kidney dysfunction.

Considering the impact of renal function on tumor marker concentrations, we analysed the changes in commonly used tumor markers in patients with CKD. We observed that the concentrations of these tumor markers, including CEA, CA125, CA153, CA199, proGRP, and HE4, increased in patients with CKD, except for CA 72-4. The results are similar to those presented in previous reports (*Miao et al., 2022*), although no statistically significant difference in CA72-4 was observed between patients with CKD and healthy individuals in our new findings. These results indicate that CKD has no effect on the serum concentration of CA72-4. The application of CA72-4 in tumor monitoring may also be applicable to tumor patients with CKD. CA72-4 is a marker associated with adenoepithelial-derived tumors (*Xu et al., 2021*). The expression concentration of CA72-4 is low in healthy individuals but high in gastric, colon (*Cho et al., 2019*), pancreatic, breast (*Li, Men & Zhang, 2020*) and lung cancers (*Mariampillai et al., 2017*). CA72-4 expression can also serve as an indicator for the diagnosis and evaluation of the therapeutic effects of the aforementioned tumors, particularly in the digestive system (*Mariampillai et al., 2017*). Previous studies have reported that the positive rate of CA72-4 expression in pancreatic cancer reached 82% (*Mariampillai et al., 2017*). Considering that CKD has little effect on CA72-4 concentrations, its clinical application may also be relevant to patients with CKD having tumors.

Thereafter, we further analysed the relationship between relevant biomarkers and CKD staging. The results suggested that there is a statistically significant difference between proGRP and HE4 in patients with different stages of CKD. The correlation analysis between different tumor marker concentrations and CKD staging showed that proGRP and HE4 were positively correlated with the stages of chronic renal insufficiency. Furthermore, the concentrations of HE4 and proGRP increased even in the early stages of renal function injury (CKD1–2). The clinical significance of this result is crucial. In particular, our study showed that the age of patients in the CKD group was higher than that of the control group, and the difference was statistically significant. Meanwhile, the tumor markers and creatinine concentrations of the patients increased along with eGFR. The early symptoms

of CKD are not apparent. When typical symptoms appear, renal function has already significantly reduced, and combined with damage to other organs, this can be extremely harmful to the body. Therefore, when elevated concentrations of relevant biomarkers are detected during routine health screenings, clinicians should be prompted to consider the possibility of underlying CKD in the patient.

Our study demonstrated that the biomarkers CEA, CA15-3, CA19-9, NSE, proGRP, and HE4 were found to be elevated in patients with CKD who presented with normal creatinine concentrations, consistent with findings from previous studies (*Mikkelsen et al., 2017*; *Huang et al., 2023*). Among these biomarkers, proGRP and HE4 showed comparatively superior diagnostic performance. Several factors may account for this phenomenon, with the most significant likely being the inflammatory state observed in patients with CKD. Prior research has reported elevated concentrations of inflammatory markers, such as interleukin-6 (IL-6) or tumor necrosis factor-alpha (TNF-$\alpha$), in individuals with CKD (*Cheung, Paik & Mak, 2010*; *Zhang, Widdop & Ricardo, 2024*). Meanwhile, numerous studies have proposed a potential association between elevated tumor markers and inflammation. For example, HE4 has been associated with inflammatory conditions in patients with idiopathic pulmonary fibrosis (IPH) (*Tian et al., 2024*) and asthma (*Zhang et al., 2024*). Similarly, the atypical elevation of tumor markers in CKD patients may be attributed to increased concentrations of inflammatory cytokines.

Additionally, some tumor markers, such as HE4 and proGRP, may be involved in the progression of patients with CKD. Consistent with our findings, previous studies have shown that HE4 can aid in identifying SLE patients with lupus nephritis (*Yang et al., 2016*; *Ren et al., 2018*). Besides, *Li, Zhong & Yang (2024)* found elevated HE4 concentrations in a lupus nephritis mouse model, and HE4 knockdown *via* an adeno-associated virus alleviated renal fibrosis progression by regulating the C3/MMPs/prss axis. Another study reported increased proGRP concentrations in plasma from hyperuricemic nephropathy patients and mouse models (*Sun et al., 2023*). The GRP/GRPR signaling pathway promotes hyperuricemia-induced renal inflammation and fibrosis progression through an ABCG2-dependent mechanism. Treatment with the proGRP inhibitor RH-1402 improved renal function and mitigated fibrosis in hyperuricemic nephropathy mouse models. These studies may elucidate certain aspects of the atypical elevation of tumor markers observed in patients with CKD who exhibit normal creatinine.

CKD has become a global public health problem owing to its high morbidity and mortality and the huge economic burden. The early detection and treatment of CKD and the delayed occurrence of serious complications are important measures to reduce the socio-economic burden of CKD. Our study revealed abnormalities in HE4 and proGRP during the early stages of renal insufficiency, which can serve as potential biomarkers for evaluating early renal dysfunction. HE4 and proGRP also showed good predictive value in patients with early renal function having normal serum creatinine. This was conducive to the diagnosis and timely treatment of early renal disease, consistent with the findings of *Miao et al. (2022)*. In combination with kidney disease, the clinical significance of elevated tumor markers should be carefully interpreted. These biomarkers have been linked with other diseases besides tumor, for example CA125 and heart failure. This study is an attempt

to explore the clinical potential of these biomarkers. First, early diagnosis of CKD is challenging due to the asymptomatic nature of the disease in its initial stages. For patients undergoing routine health examinations, elevated tumor marker concentrations, after excluding conditions such as malignancies, should prompt consideration of potential CKD. Second, during follow-up of CKD patients with normal creatinine, abnormal elevations in tumor markers should warrant enhanced evaluation and monitoring of renal function, such as glomerular filtration rate. Finally, in advanced CKD patients, abnormally elevated tumor markers may primarily reflect the underlying CKD rather than malignancy, reducing unnecessary concern about tumor-related causes. Therefore, it is of great significance to identify elevated tumor markers associated with CKD.

Our study had some limitations. First, the number of people included in the disease population was small. This is due to the fact that normal creatinine concentrations can obscure early-stage CKD, rendering these patients difficult to diagnose and subsequently enroll in research studies. In addition, there were differences in the age and sex ratios between the control and the CKD groups, which may have led to some degree of bias in the analysis. In the future, we plan to include a larger population for further analysis and verify the accuracy of the results obtained in this study. Additionally, we plan to explore the potential of these biomarkers not only in predicting renal dysfunction but also in assessing CKD progression and their association with cardiovascular events.

## CONCLUSIONS

Patients with CKD have higher tumor markers, some of which are helpful for the early diagnosis of renal impairment.

## ACKNOWLEDGEMENTS

We thank all patients, their families, and the physicians who participated in this study.

### Funding

This work was supported by the 2021 Fujian Province Young and Middle-aged Teachers' Educational Research Project (JAT210115). The funders had no role in study design, data collection and analysis, decision to publish, or preparation of the manuscript.

### Grant Disclosures

The following grant information was disclosed by the authors:
2021 Fujian Province Young and Middle-aged Teachers' Educational Research Project: JAT210115.

### Competing Interests

The authors declare there are no competing interests.

## Author Contributions

- Liuxia You conceived and designed the experiments, performed the experiments, prepared figures and/or tables, authored or reviewed drafts of the article, and approved the final draft.
- Zhongying Xu analyzed the data, prepared figures and/or tables, authored or reviewed drafts of the article, and approved the final draft.
- Qiaoling Liu performed the experiments, prepared figures and/or tables, and approved the final draft.
- Guoming Jin analyzed the data, prepared figures and/or tables, and approved the final draft.
- Changting Wang conceived and designed the experiments, authored or reviewed drafts of the article, and approved the final draft.

## Human Ethics

The following information was supplied relating to ethical approvals (i.e., approving body and any reference numbers):

The Second Affiliated Hospital of Fujian Medical University granted ethical approval to carry out the study within tis facilities.

## Data Availability

The raw measurements are available in the Supplementary Files.

## Supplemental Information

Supplemental information for this article can be found online at http://dx.doi.org/10.7717/peerj.19240#supplemental-information.

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
