# Peer review of "Tumor marker elevations in chronic kidney disease: a retrospective study"

_PeerJ, doi:10.7717/peerj.19240_

## Round 0.1 · original submission · Major Revisions

Upon review of your manuscript the referees have identified several points for improvement. In particular, it is very important to address the validation of predictive markers against known markers of renal dysfunction and any inference of mechanisms involved.

Reviewer 1 ·

Basic reporting

This article is well organized, with professional English throughout. The literature references is reasonable and sufficient. However, there are some spelling mistakes, such as"CDK" in figure titles.

Experimental design

This is an original primary research, with a well defined research question. However, the sample number of the study is quite small.

Validity of the findings

The authors showed that the tumors markers, including CEA, CA15-3, CA19-9, NSE, proGRP and HE4 were elevated in patients with CKD and " predictive value" in diagnosis of renal dysfunction. This finding is in doubt. We usually used serum creatinine or cystatin C to find renal dysfunction. If the authors intended to validate the predictive value of tumor markers, they should test the consistency of tumor markers with creatinine and cystatin C, and for the best, explore the biological relationship between tumor markers and renal dysfunction.

Additional comments

This article is well organized, with professional English throughout. The literature references is reasonable and sufficient. However, there are some spelling mistakes, such as"CDK" in figure titles. However, the sample number of the study is quite small. The sample number of the study is quite small. And the most importantly, the authors found " predictive value" of tumor markers in diagnosis of renal dysfunction. This finding is in doubt. We usually used serum creatinine or cystatin C to find renal dysfunction. If the authors intended to validate the predictive value of tumor markers, they should test the consistency of tumor markers with creatinine and cystatin C, and for the best, explore the biological relationship between tumor markers and renal dysfunction.

Reviewer 2 ·

Basic reporting

No comment

Experimental design

No comment

Validity of the findings

The study seems to be exploring an established area of research. It would be beneficial to highlight any unique aspects or novel findings. While the study identifies a correlation between tumor markers and renal dysfunction, it does not delve into the underlying mechanisms.

Additional comments

Specific suggestions: Discuss the potential role of inflammation in the elevation of tumor markers in CKD. Explore the impact of oxidative stress on tumor marker expression and renal function. Consider the influence of hormonal imbalances, such as alterations in sex hormone levels, on tumor marker expression. Emphasize the potential of tumor markers as early biomarkers for renal dysfunction. Discuss the possibility of using tumor markers to stratify patients for more aggressive management.

---

## Round 0.2 · Minor Revisions

The reviewer has recommended minor revisions to your manuscript, please take care to address the comments. In particular, clarification on the statistical analyses in of high importance.

Reviewer 2 ·

Basic reporting

Reviewer Report: "Role of Tumor Markers in Chronic Kidney Disease: A Retrospective Study"
Title: The study investigates the role of tumor markers (CA125, CA15-3, CA19-9, CA72-4, proGRP, and HE4) in chronic kidney disease (CKD), comparing the levels of these markers in CKD patients to healthy controls and evaluating their potential use for early detection of renal dysfunction. The investigation of tumor markers in CKD is timely and relevant, especially considering the increasing prevalence of CKD worldwide. The study included a reasonable number of participants (522), making the results robust and generalizable. The study provides valuable insights into the role of tumor markers for detecting early renal dysfunction, particularly in CKD patients with normal serum creatinine levels, an area with limited prior research. Appropriate statistical tests were used, including Pearson correlation analysis and ROC curve analysis, which provide confidence in the validity of the findings.

Although the findings suggest that these biomarkers may aid early detection of CKD, the clinical utility in routine practice remains uncertain. Further research is needed to confirm whether these markers could be integrated into clinical guidelines or used as part of a broader diagnostic panel.
This study provides compelling evidence that certain tumor markers (specifically HE4 and proGRP) could play a role in the early detection of renal dysfunction in CKD patients, even in the absence of overt symptoms or abnormal creatinine levels. The findings are promising, but further research, including prospective studies with larger and more diverse populations, is necessary to validate the clinical utility of these markers. The manuscript is well-written, but addressing the identified weaknesses will enhance its impact and scientific rigor.

Experimental design

The study investigates the role of tumor markers (CA125, CA15-3, CA19-9, CA72-4, proGRP, and HE4) in chronic kidney disease (CKD), comparing the levels of these markers in CKD patients to healthy controls and evaluating their potential use for early detection of renal dysfunction. The investigation of tumor markers in CKD is timely and relevant, especially considering the increasing prevalence of CKD worldwide. The study included a reasonable number of participants (522), making the results robust and generalizable.

Validity of the findings

The study provides valuable insights into the role of tumor markers for detecting early renal dysfunction, particularly in CKD patients with normal serum creatinine levels, an area with limited prior research. Appropriate statistical tests were used, including Pearson correlation analysis and ROC curve analysis, which provide confidence in the validity of the findings.

Additional comments

Additional Comments:
• Literature Review: The manuscript could benefit from a more comprehensive review of the existing literature on tumor markers in CKD. This would provide a stronger contextual foundation for the study's findings and demonstrate how they contribute to the existing body of knowledge.
• Future Directions: The authors should consider further validating these findings in larger, multi-center studies and exploring the biomarkers' potential for predicting not only renal dysfunction but also CKD progression and associated cardiovascular events.
• Clarification of Statistical Analysis: It would be helpful for the authors to provide more detail on how they handled potential confounding factors in their statistical models, especially given the differences between groups.

---

## Round 0.3 · Minor Revisions

Dear authors,
Thank you for your resubmission.Before we proceed to the production phase, there are several minor issues that need to be addressed to ensure the clarity and accuracy of your manuscript.
First, the legends for the tables and figures need to be revised for better clarity and informativeness. For instance, the legend for "Table 3: The value of tumor markers to distinguish patients with CKD from healthy individuals" is unclear—specifically, what exactly does "the value of tumor markers" refer to, and how is it measured? Similarly, in "Table 2: The level of biomarkers in control group and different stages of CKD patients," the term "level" should be defined—does it refer to concentration, and if so, in what volume of sample? Only the legend for Figure 1 is currently clear and acceptable. Please make similar improvements for the other figures and tables.
Additionally, the way data is presented in the manuscript is somewhat unconventional, especially in relation to biomarker analyses. For example, AUC values are presented in a table rather than within the ROC curves, which is more typical for this field. It is crucial to ensure that the data presented is accurate, particularly in the context of biomarkers, where precision is key.
There are also inconsistencies in the reporting of data. In "Table 1: Clinical characteristics of control and CKD groups," the reporting of age in the control group as "38.19 (32.00, 45.00)" is confusing, as are similar reports for other variables (e.g., Cr (umol/L) 70.06 (58.22, 81.85), Ur 11.43 (5.69, 13.85)). These discrepancies raise concerns about the reliability of the statistical analysis, especially when significant p-values are reported across most analyses. I encourage you to review the consistency of your data presentation, particularly considering the different group sizes.
There is also some ambiguity regarding the design of the study, particularly with the control and experimental groups. It is important to ensure that the study design is presented clearly, especially in terms of group selection and experimental protocols.
Furthermore, the methods section should be more detailed. The authors should provide full information about the kits and reagents used, including reference numbers for products such as the "commercial chemiluminescent microparticle immunoassay." Clear and transparent reporting of methods and materials is essential for reproducibility.
The statement "analyzed the efficacy of tumor markers in distinguishing CKD patients from 90 healthy controls" in the methods section is somewhat ambiguous. It would be clearer to specify that the study evaluated the levels or concentrations of specific proteins or enzymes in plasma, or clarify this point further.
Please also ensure that any acronyms used in the manuscript are defined when they first appear. Additionally, the title may need revision to better reflect the scope and conclusions of your study.
Lastly, it would be helpful to provide additional context regarding why the study focuses exclusively on these markers, particularly given the confounding complexity of urinary physiology. How is this approach relevant, and what are the clinical implications of your findings? These clarifications are necessary to ensure the manuscript meets scientific standards and provides a comprehensive understanding of the study's relevance.

---

## Round 0.4 · Minor Revisions

Dear authors,
i appreciate your work and revisions. However, i think some things have been maybe overlooked:
- in the methods, you report using the ARCHITECT (from Abbott labs), but it is an immunoassay analyzer, therefore you should actually mention ARCHITECT ???? (model) immunoassay analyzer, and then which "chemiluminescent microparticle immunoassays" - ref numbers of the different assays, clearly registered.
- attention to the term "level" (should be clearly defined somewhere in the manuscript and i could not find such)
- relevant information already reported in the rebuttal should also have been incorporated in to the manuscript; for example "To ensure the accuracy of data presentation, we conducted normality tests on the relevant variables before reporting. We presented the data as mean ± standard deviation (SD) for variables that followed a normal distribution. For non-normally distributed variables, we used the median and interquartile range (IQR) to describe the data. This approach allows for a more accurate reflection of the data distribution and ensures consistency in our reporting.
Moreover, in our statistical analysis, non-normally distributed data were analyzed using non-parametric tests, as appropriate, to account for the distribution characteristics and maintain the robustness of our conclusions. "
and
"Several studies have assessed changes in tumor markers in specific populations, including patients with CKD. For instance, a study by Tong et al. (Tong et al., 2013) evaluated tumor markers such as CEA, NSE, AFP, CY211, and SCC in CKD patients. The study included 539 non-dialysis CKD patients and 223 healthy controls. It found that the false-positive rates of CEA, SCC, and CY211 were significantly higher in CKD patients compared to healthy controls. There was no significant difference between CKD patients and healthy controls for AFP and NSE. Additionally, the false-positive rates of Cyfra21-1 and SCC in CKD patients increased progressively with the worsening of CKD. Another study by Rani et al. (Rani et al., 2019) assessed changes in CA15-3, CEA, CA 19-9, and human HCG in advanced CKD patients compared to healthy controls. The results indicated that the levels of these tumor markers were significantly elevated in CKD patients compared to controls, but no significant difference was observed in CKD stage 4 and 5 patients. Furthermore, Huang et al. (Huang et al., 2023) evaluated the correlation of serum Galectin-3 and CA15-3 concentrations with kidney function, revealing a significant negative correlation between Galectin-3, CA15-3, and eGFR. This relationship was more pronounced in overweight and obese individuals.
These studies suggest that tumor marker levels in CKD populations may be influenced by kidney function. However, the existing literature has not further evaluated the role of abnormal tumor marker levels in the differentiation of CKD, especially in the identification of early-stage CKD with kidney dysfunction.
Considering the impact of renal function on tumor marker levels, we analysed the changes in commonly used tumor markers in patients with CKD. We observed that the levels of these tumor markers, including CEA, CA125, CA153, CA199, proGRP, and HE4, increased in patients with CKD, except for CA 72-4. The results are similar to those presented in previous reports(Miao et al., 2022) , although no statistically significant difference in CA72-4 was observed between patients with CKD and healthy individuals in our new findings. These results indicate that CKD has no effect on the serum level of CA72-4. The application of CA72-4 in tumor monitoring may also be applicable to tumor patients with CKD. CA72-4 is a marker associated with adenoepithelial-derived tumors(Xu et al., 2021). The expression level of CA72-4 is low in healthy individuals but high in gastric, colon(Cho et al., 2019), pancreatic, breast(Li et al., 2020) and lung cancers(Mariampillai et al., 2017). Considering that CKD has little effect on CA72-4 levels, its clinical application may also be relevant to patients with CKD having tumors.
In combination with kidney disease, the clinical significance of elevated tumor markers should be carefully interpreted. These biomarkers have been linked with other diseases besides tumor, for example CA125 and heart failure. This study is an attempt to explore the clinical potential of these biomarkers. First, early diagnosis of CKD is challenging due to the asymptomatic nature of the disease in its initial stages. For patients undergoing routine health examinations, elevated tumor marker levels, after excluding conditions such as malignancies, should prompt consideration of potential CKD. Second, during follow-up of CKD patients with normal creatinine levels, abnormal elevations in tumor markers should warrant enhanced evaluation and monitoring of renal function, such as glomerular filtration rate. Finally, in advanced CKD patients, abnormally elevated tumor markers may primarily reflect the underlying CKD rather than malignancy, reducing unnecessary concern about tumor-related causes. Therefore, it is of great significance to identify elevated tumor markers associated with CKD.
"....
while not exactly by these words, but it still should be in the manuscript
- something happened with your references list as it disappeared and was substituted only by the library file name

---

## Round 0.5 · accepted · Accept

Dear authors,

I am accepting your manuscript for publication. Please, be mindful and throughout in your proofreading. All the best!